Shedding light on the Ophel biome: the trans-Tethyan phylogeography of the sulfide shrimp Tethysbaena (Peracarida: Thermosbaenacea) in the Levant

Guy-Haim Tamar tamar.guy-haim@ocean.org.il 1
Kolodny Oren 2
Frumkin Amos 3
Achituv Yair 4
Velasquez Ximena 1
Morov Arseniy R. 1
1 National Institute of Oceanography, Israel Oceanographic and Limnological Research , Haifa , Israel
2 Department of Ecology, Evolution, and Behavior, Institute for Life Sciences, The Hebrew University of Jerusalem , Jerusalem , Israel
3 Institute of Earth Sciences, The Hebrew University of Jerusalem , Jerusalem , Israel
4 The Mina and Everard Goodman Faculty of Life Sciences, Bar Ilan University , Ramat-Gan , Israel
Langille Barbara
Electronic publication date: 2023 Dec 20
Publication date: 2023
Volume: 11
Electronic Location ID: e16690
Received 2023 Aug 17; Accepted 2023 Nov 27
Copyright: ©2023 Guy-Haim et al.
Copyright year: 2023
Copyright holder: Guy-Haim et al.
License: This is an open access article distributed under the terms of the Creative Commons Attribution License, which permits unrestricted use, distribution, reproduction and adaptation in any medium and for any purpose provided that it is properly attributed. For attribution, the original author(s), title, publication source (PeerJ) and either DOI or URL of the article must be cited.
License URL: https://creativecommons.org/licenses/by/4.0/

Keywords: Ayyalon cave, Dead Sea-Jordan Rift Valley, Oligocene-Miocene marine transgressions, 28S rRNA, COI mtDNA, Phylogeny, Relics, Stygofauna, Tethys Sea, Ophel paradigm

Funding: The authors received no funding for this work.

==============================
Background

Tethysbaena are small peracarid crustaceans inhabiting extreme environments such as subterranean lakes and thermal springs, represented by endemic species found around the ancient Tethys, including the Mediterranean, Arabian Sea, Mid-East Atlantic, and the Caribbean Sea. Two Tethysbaena species are known from the Levant: T. relicta, found along the Dead Sea-Jordan Rift Valley, and T. ophelicola, found in the Ayyalon cave complex in the Israeli coastal plain, both belonging to the same species-group based on morphological cladistics. Along the biospeleological research of the Levantine subterranean fauna, three biogeographic hypotheses determining their origins were proposed: (1) Pliocenic transgression, (2) Mid-late Miocenic transgression, and (3) The Ophel Paradigm, according to which these are inhabitants of a chemosynthetic biome as old as the Cambrian.

Methods

Tethysbaena specimens of the two Levantine species were collected from subterranean groundwaters. We used the mitochondrial cytochrome c oxidase subunit I (COI) gene and the nuclear ribosomal 28S (28S rRNA) gene to establish the phylogeny of the Levantine Tethysbaena species, and applied a molecular clock approach for inferring their divergence times.

Results

Contrary to the morphological cladistic-based classification, we found that T. relicta shares an ancestor with Tethysbaena species from Oman and the Dominican Republic, whereas the circum-Mediterranean species (including T. ophelicola) share another ancestor. The mean age of the node linking T. relicta from the Dead Sea-Jordan Rift Valley and Tethysbaena from Oman was 20.13 MYA. The mean estimate for the divergence of T. ophelicola from the Mediterranean Tethysbaena clade dated to 9.46 MYA.

Conclusions

Our results indicate a two-stage colonization of Tethysbaena in the Levant: a late Oligocene transgression, through a marine gulf extending from the Arabian Sea, leading to the colonization of T. relicta in the Dead Sea-Jordan Rift Valley, whereas T. ophelicola, originating from the Mesogean ancestor, inhabited anchialine caves in the coastal plain of Israel during the Mid-Miocene.

Introduction

Groundwater fauna (stygofauna) is characterized by short-range endemism and high species crypticity (Eme et al., 2013; Zagmajster et al., 2014). The unique suite of troglomorphic traits (e.g., loss of pigment, reduced eyes) characterizing stygobionts often hinders distributional studies due to the highly convergent characteristics that can obscure taxonomic relationships (Juan et al., 2010; Porter, 2007). As a result, molecular phylogenetic tools have been extensively used over the last two decades to infer stygofauna biogeographies and the underlying processes shaping them (e.g., Abrams et al., 2019; Asmyhr et al., 2014; Bauzà-Ribot et al., 2012; Bradford et al., 2010; Cánovas et al., 2016; Cooper et al., 2023; Finston et al., 2004; Guy-Haim et al., 2018; Jaume, 2008; Jurado-Rivera et al., 2017; Marin, Krylenko & Palatov, 2021; Matthews et al., 2020).

Thermosbaenacea is a small order of peracarid crustaceans comprising 36 unique and highly specialized species adapted to extreme aquatic environments, including spring-fed subterranean lakes and thermal springs, with their core populations found deep underground in the inaccessible phreatic waters (Por, 2014; Wagner, 1994; WoRMS, 2023). Anoxic, sulfide-rich environments are favorable to Thermosbaenacea—often feeding on bacterial mats formed by sulfide-oxidizing bacteria—thus termed “sulfide shrimp” by Por (2014). Based on their distribution, it was assumed that the ancestral habitat of the thermosbaenaceans is the Tethys—an ancient ocean that was situated between Laurasia to the north and Gondwana to the south and existed from the late Paleozoic to the early Cenozoic eras (ca. 250–5 MYA). Nowadays, Thermosbaenacea representatives, largely relict and geographically isolated species, are found around the Mediterranean, the Arabian Sea, the Indian Ocean, Mid-East Atlantic, and the Caribbean Sea (Hou & Li, 2018; Wagner, 1994). Among thermosbaenaceans, Tethysbaena (family: Monodellidae) is the most speciose and widespread genus, comprising 27 species in seven species-groups (Wagner, 1994; Wagner & Bou, 2021). Only a few of the Tethysbaena species-groups were analyzed and supported by molecular phylogenetic tools (Cánovas et al., 2016; Wagner & Chevaldonné, 2020).

Two species of Tethysbaena are known from Israel: T. relicta (Por, 1962) (formerly Monodella relicta) and T. ophelicola (Wagner, 2012). Initially, fragments of T. relicta were found in the hot sulfide-rich spring Hamei Zohar by the Dead Sea in Israel (Por, 1962). Later, a few specimens of the same species were occasionally found in the thermohaline spring En-Nur, on Lake Kinneret shore, a few hundred kilometers to the north (Dimentman & Por, 1991), thus inferring that T. relicta inhabits the whole groundwater system of the Dead Sea-Jordan Rift Valley aquifer. T. ophelicola was found in the karstic underground basin near Ramla, named Ayyalon-Nesher-Ramla complex (Por, 2014; Por et al., 2013; Wagner, 2012), 60 km west of the Dead Sea-Jordan Rift Valley, beyond the water divide of Israel, thus belonging to a different watershed system. Similarly to other Thermosbaenacea species, the Levantine Tethysbaena inhabit environments characterized by high temperatures and sulfide-rich waters—31 °C in Hamei-Zohar (Por, 1962), 30–31 °C with 60–190 ppm sulfate in En-Nur (Beeri-Shlevin et al., 2022; Tsurnamal, 1978), and 28.8–29.6 °C with 160–305 ppm sulfate concentrations in the Ayyalon cave complex (Langford et al., 2022; Por et al., 2013).

Based on synapomorphies of the antennular inner flagellum and maxilliped macrosetae (Wagner, 1994), it was hypothesized that together with other closely allied species (one species from Somalia (Chelazzi & Messana, 1982), four species from Oman (Wagner, 2020), one species from Yemen (Wagner & Van Damme 2021)), T. relicta and T. ophelicola form the “T. relicta-group” (Wagner, 2012), suggesting a recent common ancestor. An alternate hypothesis can be drawn from the phylogenetic analysis of the prawn Typhlocaris (Guy-Haim et al., 2018), preying on Tethysbaena in Ayyalon and En-Nur (Tsurnamal, 1978; Tsurnamal, 2008; Tsurnamal & Por, 1971; Wagner, 2012). Four Typhlocaris species are known, two of which co-occur with Tethysbaena in Israel: Ty. galilea inhabiting En-Nur spring (Calman, 1909; Tsurnamal, 1978) and Ty. ayyaloni from the Ayyalon cave (Tsurnamal, 2008). The two additional Typhlocaris species are Ty. salentina from Southeastern Italy (Caroli, 1923; Froglia & Ungaro, 2001) and Ty. lethaea from Libya (Parisi, 1921). The molecular phylogeny of Typhlocaris species showed that Ty. ayyaloni (Israel) and Ty. salentina (Italy) are more closely related to each other than either of them is to Ty. galilea (Israel) (Guy-Haim et al., 2018). Accordingly, we can hypothesize a similar phylogeographic pattern of the Levantine Tethysbaena, where T. ophelicola would be more closely related to the Mediterranean species (“T. argentarii-group”) than to T. relicta.

Along the biospeleological research of the Thermosbaenacea and other phyla of subterranean crustacean fauna represented in the Dead Sea Rift Valley (Syncarida, the families Bogidiellidae, Typhlocarididae, Niphargidae, Sphaeromatidae, and Cirolanidae), three paradigms have been proposed to explain their origins. Accordingly, Tethysbaena colonization could be attributed to: (1) Pliocene pre-glacial (Piacenzian, 3.600–2.588 MYA) marine transgression (Por, 1963), penetrating the Dead Sea Rift Valley from the Mediterranean; (2) mid-Miocene Tethys transgression into the Levant (16–10 MYA) (Dimentman & Por, 1991; Por, 1987); and (3) The Ophel Paradigm that offered a conceptual framework, within which these styobionts are inhabitants of the ancient chemosynthetic Ophel biome, dating back at least to the Cambrian (>485 MYA) (Por, 2011).

Using a molecular clock approach, Guy-Haim et al. (2018) estimated the divergence time of the Typhlocaris species. They based their analysis on a calibration node inferred from a regional geological event—the end of the marine connection between the Mediterranean Sea and the Dead Sea-Jordan Rift Valley, marked by the top of Bira formation, dated to 7 MYA (Rozenbaum et al., 2016), separating Ty. galilea and the Typhlocaris ancestor. The inferred divergence time of Ty. ayyaloni and Ty. salentina was 5.7 (4.4–6.9) MYA, at the time of the Messinian Salinity Crisis (5.96–5.33 MYA), when the Mediterranean Sea desiccated (Garcia-Castellanos & Villaseñor, 2011; Gargani & Rigollet, 2007) and lost almost all its Miocene tropical fauna (Por, 1987; Por, 1989). It is therefore an open question as to whether the same vicariant events have shaped the biogeographies of both predator (Typhlocaris) and prey (Tethysbaena) subterranean crustaceans.

The main objectives of our study were to (1) reveal the phylogenetic relatedness of the Levantine Tethysbaena species, and use these patterns to (2) infer the geological and evolutionary processes that have shaped their divergence patterns. To do so, we used the mitochondrial COI and nuclear 28S rRNA genes to construct the phylogeny of T. relicta and T. ophelicola and estimated their divergence ages by applying molecular clock approach.

Materials & Methods

Sampling sites, specimen collection and identification

Specimens of T. ophelicola were collected by a hand pump from the surface water of the inner pool (maximal depth ca. 2 m) of the Levana cave (31.9223°N, 34.8942°E), part of the Ayyalon-Nesher-Ramla complex (Fet et al., 2017) (Fig. 1).

Figure 1 Tethysbaena distribution and habitats.

(A) Global Tethysbaena distribution. The species included in the phylogenetic analysis are presented in colored circles. Other Tethysbaena species are presented in black. Based on documented records in Wagner (1994), Wagner (2012), Wagner (2020), Cánovas et al. (2016), Wagner & Chevaldonné (2020) and Wagner & Bou (2021). (B) Levantine distribution of T. ophelicola and T. relicta. JRV –Jordan Rift Valley. (C–D) Tethysbaena Levantine habitats. (C) An artificial tunnel near the Dead Sea, Israel. (D) Levana (Ayyalon) cave, Israel.

Specimens of T. relicta were collected by a hand pump from the overlying water near the bottom of an artificial tunnel (maximal depth ca. 0.6 m) near the Dead Sea Shore penetrating the Judea Group aquifer, 6.5 km north of Hamei-Zohar (31.2232°N, 35.3547°E) (Fig. 1). The locus typicus of T. relicta, the thermal spring of Hamei-Zohar (Por, 1962), is no longer accessible since the 1970s, as hotels were built on the spring area. Nonetheless, since Dimentman & Por (1991) found scattered T. relicta specimens in En-Nur, 150 km north of the type locality Hamei-Zohar, we assumed that the new Tethysbaena record, located along the Dead Sea Jordan Rift, is of T. relicta, and validated it using morphological characterization.

Part of the collected specimens was preserved in 70% ethanol and the other in absolute ethanol for morphological and molecular analyses, respectively. Species identification of T. ophelicola and T. relicta was performed using a stereomicroscope (SZX16, Olympus, Japan) following the identification keys in Por (1962), Wagner (1994) and Wagner (2012).

DNA extraction, amplification and sequencing

Cánovas et al. (2016) used both COI and nuclear 28S rRNA genes to assess the genetic population structure of the anchialine T. scabra in the Balearic Islands, and found that the 28S rRNA gene showed low genetic variation resulting in a poorly resolved phylogenetic tree, and they, therefore, based their phylogenetic reconstruction and divergence time estimations on the COI gene only. Nonetheless, since both COI and 28S rRNA sequences of Tethysbaena are available in NCBI GenBank (https://www.ncbi.nlm.nih.gov/genbank/), we have used these two genes in our analysis.

Total genomic DNA was extracted from each individual using the DNeasy Blood and Tissue Kit (QIAGEN, Hilden, Germany) according to the manufacturer’s specifications. Following the DNA extraction, the COI gene was amplified using PCR with universal primers LCO1490 and HCO2198 (Folmer et al., 1994), and 28S rRNA gene was amplified using the primers 28S-1274 5′-GACCCGTCTTGAAACACGGA-3′(Markmann & Tautz, 2005) and 28S-D6br 5′-CACACGAAACCCTTCTCCAC-3′(Omilian & Taylor, 2001), following Cánovas et al. (2016).

Reaction conditions for COI gene amplification were as follows: 94 °C for 2 min, followed by 5 cycles of 94 °C for 40 s, 45 °C for 40 s, and 72 °C for 1 min, and followed by 30 cycles of 94 °C for 40 s, 51 °C for 40 s, and 72 °C for 1 min, and a final elongation step of 72 °C for 10 min. Reaction conditions for 28S rRNA gene amplification were: 95 °C for 5 min, followed by 30 cycles of 92 °C for 25 s, 53 °C for 90 s, and 72 °C for 25 s, and a final elongation step of 72 °C for 7 min. Obtained PCR products were purified and sequenced by Hylabs (Rehovot, Israel).

Phylogenetic analysis

A total of 22 COI sequences of Tethysbaena were analyzed, including T. ophelicola (n = 3) and T. relicta (n = 3) obtained in this study. Additional sequences of T. scabra (Balearic Islands, n = 5), T. argentarii (Italy, n = 2), T. ledoyeri (France, n = 2), T. atlantomaroccana (Morocco, n = 1), and further sequences of Tethysbaena sp., unidentified to the species level, from Oman (n = 2), Morocco (n = 3) and the Dominican Republic (n = 1), were obtained from NCBI GenBank and the European Nucleotide Archive (https://www.ebi.ac.uk/ena/browser/home). The thermosbaenacean Halosbaena tulki was chosen as an outgroup following Page et al. (2016) and used as a root node in the phylogenetic analysis. All specimens, collection sites, accession numbers, and related references are summarized in Table 1.

Table 1 COI sequences of Tethysbaena and outgroup included in the phylogenetic analysis.

	Species	Accession number	Locality	Reference	
1	Tethysbaena relicta	OR189199.1	Dead Sea tunnel, Israel	This study	
2	Tethysbaena relicta	OR189200.1	Dead Sea tunnel, Israel	This study	
3	Tethysbaena relicta	OR189201.1	Dead Sea tunnel, Israel	This study	
4	Tethysbaena ophelicola	OR189202.1	Levana cave, Israel	This study	
5	Tethysbaena ophelicola	OR189203.1	Levana cave, Israel	This study	
6	Tethysbaena ophelicola	OR189204.1	Levana cave, Israel	This study	
7	Tethysbaena ledoyeri	QLI41807.1	Southern France	Wagner & Chevaldonné (2020)	
8	Tethysbaena ledoyeri	QLI41808.1	Southern France	Wagner & Chevaldonné (2020)	
9	Tethysbaena argentarii	LN899289.1	Monte Argentario, Italy	Cánovas et al. (2016)	
10	Tethysbaena argentarii	LN899302.1	Monte Argentario, Italy	Cánovas et al. (2016)	
11	Tethysbaena scabra	LN899332.1	Balearic Islands	Cánovas et al. (2016)	
12	Tethysbaena scabra	LN899343.1	Balearic Islands	Cánovas et al. (2016)	
13	Tethysbaena scabra	LN899375.1	Balearic Islands	Cánovas et al. (2016)	
14	Tethysbaena scabra	LN899402.1	Balearic Islands	Cánovas et al. (2016)	
15	Tethysbaena scabra	LN899310.1	Balearic Islands	Cánovas et al. (2016)	
16	Tethysbaena atlantomaroccana	LN899421.1	Marrakech, Morocco	Cánovas et al. (2016)	
17	Tethysbaena sp. 1	LN899418.1	Dhofar coast, Oman	Cánovas et al. (2016)	
18	Tethysbaena sp. 2	LN899419.1	Dhofar coast, Oman	Cánovas et al. (2016)	
19	Tethysbaena sp. 3	LN899420.1	Southwest Dominican Republic	Cánovas et al. (2016)	
20	Tethysbaena sp. 4	LN899422.1	Tasla, Morocco	Cánovas et al. (2016)	
21	Tethysbaena sp. 5	LN899423.1	Tasla, Morocco	Cánovas et al. (2016)	
22	Tethysbaena sp. 6	LN899424.1	Lamkedmya, Morocco	Cánovas et al. (2016)	
23	Halosbaena tulki (outgroup)	KT984092.1	Australia	Page et al. (2016)	

The 28S rRNA analysis included six sequences of Tethysbaena, of which 5 obtained in this study—T. ophelicola (n = 2) and T. relicta (n = 3), and one was obtained from NCBI GenBank (T. argentarii, Italy, n = 1). The twenty-one 28S rRNA sequences of T. scabra from the Balearic Islands, and the two 28S rRNA sequences of T. argentarii produced by Cánovas et al. (2016), obtained with the same primers that we used, had a very low coverage (0–8%) when compared with the 28S rRNA sequences obtained by us or with other 28S rRNA sequences in GenBank, including those obtained by Page et al. (2016) for Halosbaena, and therefore could not be used.

Sequence alignment was conducted using ClustalW embedded in MEGA v11.0 (Tamura, Stecher & Kumar, 2021). The best-fitting substitution model was selected according to the Bayesian Information Criterion using Maximum-likelihood (ML) model selection in MEGA. ML analysis was performed using the T92+G+I model for COI, and T93+I model for 28S rRNA sequence alignments, with 1000 bootstrapping replicates. Bayesian Metropolis coupled Markov chain Monte Carlo (B-MCMC) analyses were conducted with MrBayes v3.2.7a (Ronquist et al., 2012) on XSEDE in the CIPRES v3.3 Science Gateway portal (https://www.phylo.org/portal2) with nst = 2, rates = gamma, and statefreqpr = fixed (fixedest = equal). Two independent runs of 10,000,000 generations each performed, sampling every 1,000 generations. A burn-in at 25% of the sampled trees was set for final tree production. Convergence and effective sampling of runs was assessed using Tracer v. 1.6 (Drummond & Rambaut, 2007), and the post-burnin tree samples were summarized using the sumt.

Estimation of divergence times

Molecular clock calculations for cave-dwelling species are often contentious (Page, Humphreys & Hughes, 2008). Stygobionts typically display unique evolutionary characteristics including isolation, reduced gene flow, small population sizes, and distinct selective pressures (Lefébure et al., 2017; Saclier et al., 2018). These factors can lead to deviations from a constant rate of molecular evolution among lineages, rendering a strict molecular clock assumption less realistic. Therefore, we used a relaxed molecular clock approach (Drummond et al., 2006). Cánovas et al. (2016) assessed the divergence time of the Western Mediterranean Tethysbaena, T. scabra from the Balearic Islands, and T. argentarii from Italy using the COI gene. In the absence of known Thermosbaenacea fossil record for molecular clock calibration (Jaume, 2008), they based the substitution rates on the mean rate estimated for a co-occurring anchialine stygobiont amphipod Metacrangonyx longipes, 1.32% per lineage and million years (95% CI [0.89–1.95]) (Bauzà-Ribot et al., 2012). Following Cánovas et al. (2016), we implemented this substitution rate in our COI dataset.

A relaxed-clock MCMC (Markov Chain Monte Carlo) approach using the uncorrelated log-normal model was implemented in BEAST v2.4 (Drummond & Rambaut, 2007; Suchard et al., 2018; Suchard & Rambaut, 2009). The Yule process was chosen as a speciation process due to the incomplete knowledge on Tethysbaena rates of speciation. Three independent runs, each of 50,000,000 generations, were performed, with sampling every 5000 generations. An assessment of convergence and effective sample sizes for all parameters was conducted using Tracer v1.6 (Drummond & Rambaut, 2007). The combined log files were then analyzed to ensure that the MCMC chains had run long enough to obtain valid parameter estimates. A 10% burn-in was chosen after considering convergence and effective sample sizes. The three separate runs were then combined using LogCombiner v2.5.2. Maximum clade credibility (MCC, hereafter) tree was then produced using TreeAnnotator v2.4.7 (Rambaut & Drummond, 2017). FigTree v.1.4.4 (Rambaut, 2018) was used to visualize the MCC tree and the highest posterior density (HPD, hereafter) ranges.

Results

Morphological identification

Specimens of T. relicta collected from the Dead Sea tunnel were similar to the specimens from the type locality, Hamei-Zohar thermal spring, described by Por (1962), and included males, with no ovigerous or brooding females (Fig. 2A). The average length (excluding antennae) was 2,104 ± 181 µm (n = 5, ±SD, hereafter). The following morphological features characterized the specimens as belonging to T. relicta: eight segments in the main flagellum (endopodite) of antenna 1; seven terminal plumidenticulate macrosetae were present on the maxilliped; the uropod included five medial plumose macrosetae, 11–13 plumose macrosetae in the endopodite, and 16–19 macrosetae in the second segment of the exopodite. The mean width:length ratio of the telson was 1.15.

Figure 2 Levantine Tethysbaena..

(A) Tethysbaena relicta (Por, 1962) male. (B–D) Tethysbaena ophelicola (Wagner, 2012) male (B), brooding female (C), and postmarsupial juvenile (D). The arrowhead points to the orange coloration of the gut (B), indicating the presence of sulfide-oxidizing bacteria. The scale bar denotes 200 µm in A–C and 100 µm in D.

Specimens of T. ophelicola from Levana cave were similar to the specimens from Ayyalon cave described by Wagner (2012), and included males, brooding females and postmarsupial juveniles (Figs. 2B–2D). The average length (excluding antennae) was 2,276 ± 380 µm in males (n = 5) and 2,620 ± 139 µm in females (n = 5). The following morphological features were found: seven segments in the main flagellum (endopodite) of antenna 1; seven terminal plumidenticulate macrosetae were present on the maxilliped; uropod included 4 medial plumose macrosetae and 18–22 plumose macrosetae in the endopodite and the second segment of the exopodite. The mean width:length ratio of the telson was 1.10.

Molecular phylogenetic analysis

The DNA barcode consisting of a fragment of 624–641 bp of the COI gene was sequenced from 6 specimens of T. ayyaloni and T. relicta. Sequences were deposited in NCBI GenBank under accession numbers OR189199 –OR189204. The phylogenetic analysis included 16 additional Tethysbaena sequences and one Halosbaena tulki sequence as an outgroup (Table 1). The overall COI alignment was 691 bp long, with 227 parsimonious informative sites.

The amplified 28S rRNA gene sequences of T. ayyaloni and T. relicta, deposited in NCBI GenBank under accession numbers OR790561 –OR790565, were 834–885 bp long. The phylogenetic analysis included an additional Tethysbaena sequence (T. argentarii) and three Halosbaena sequences as an outgroup (Table 2). The overall 28S rRNA alignment was 908 bp long, with 498 parsimonious informative sites.

Table 2 28S rRNA sequences of Tethysbaena and outgroup included in the phylogenetic analysis.

	Species	Accession number	Locality	Reference	
1	Tethysbaena relicta	OR790563.1	Dead Sea tunnel, Israel	This study	
2	Tethysbaena relicta	OR790564.1	Dead Sea tunnel, Israel	This study	
3	Tethysbaena relicta	OR790565.1	Dead Sea tunnel, Israel	This study	
4	Tethysbaena ophelicola	OR790561.1	Levana cave, Israel	This study	
5	Tethysbaena ophelicola	OR790562.1	Levana cave, Israel	This study	
6	Tethysbaena argentarii	DQ470654.1	Italy	Stenderup, Olesen & Glenner (2006)	
7	Halosbaena fortunata (outgroup)	KT984015.1	Spain	Page et al. (2016)	
8	Halosbaena daitoensis (outgroup)	KT984014.1	Japan	Page et al. (2016)	
9	Halosbaena tulki (outgroup)	KT984043.1	Australia	Page et al. (2016)	

The COI-based ML and Bayesian phylogenetic analyses showed similar tree topologies (Fig. 3). The Levantine Tethysbaena species from Israel present polyphyly, where T. ayyaloni is positioned within a Mediterranean clade (including T. scabra from the Balearic Islands, T. ledoyeri from Southern France and T. argentarii from Italy) with 100% bootstrapping support and 0.99 posterior probability, and T. relicta clusters with Tethysbaena sp. from Oman (100% bootstrapping support and 1.00 posterior probability), and the Dominican Republic (87%/0.83 bootstrapping support/posterior probability), forming the Arabian-Caribbean clade. The Atlantic T. atlantomaroccana is a sister taxon to the Mediterranean clade species, although with a lower support/probability. The other Moroccan Tethysbaena species from Tasla and Lamkedmya were in a more basal position but showed lower bootstrapping support (<50%).

Figure 3 Maximum-Likelihood phylogenetic tree of Tethysbaena based on the COI mitochondrial gene, using the T92+G+I substitution model.

Halosbaena tulki was used as a root node. At each node, the number on the left side of the slash indicates the percentage of ML bootstrap support (1,000 replicates), and the right number indicates the Bayesian posterior probability expressed as a decimal fraction, for nodes that received at least 50% support. The scale bar denotes the estimated number of nucleotide substitutions per site.

The 28S rRNA-based ML and Bayesian phylogenetic analyses (Fig. 4) included only a subset of the species in the COI phylogenetic analysis. Similar to the COI-based phylogenetic analysis, it showed that the Levantine T. ayyaloni is a sister taxon to the Mediterranean clade species T. argentarii from Italy (86%/1.00 bootstrapping support/posterior probability), whereas T. relicta is more distant (100%/1.00 bootstrapping support/posterior probability).

Figure 4 Maximum-Likelihood phylogenetic tree of Tethysbaena based on the 28S ribosomal gene, using the TN93+I substitution model.

Halosbaena species were used as a root node. At each node, the number on the left side of the slash indicates the percentage of ML bootstrap support (1,000 replicates), and the right number indicates the Bayesian posterior probability expressed as a decimal fraction, for nodes that received at least 50% support. The scale bar denotes the estimated number of nucleotide substitutions per site.

Divergence time estimation

Effective sample size (ESS) values were at least 436 and 356 for the posterior and prior statistics, respectively, 1,738 for the likelihood statistic, and greater than 1,400 for all MRCA times estimates, suggesting good mixing and an effective MCMC sampling of the posterior distribution.

We estimated the ages for eight nodes (Table 3, Fig. 5). The youngest node was the most recent common ancestor of T. leyoderi from Southern France and T. scabra from the Balearic Islands, which returned a mean estimate at 8.31 MYA. The next mean estimate is the divergence of T. ophelicola from the clade of T. leyoderi and T. scabra, dated to 9.46 MYA. The mean age of the most common ancestor of all Mediterranean Tethysbaena was 10.71 MYA. The most recent ancestor of the Mediterranean clade and T. atlantomaroccana from Morocco was dated to 32.41 MYA. The mean age of the node linking T. relicta from the Dead Sea-Jordan Rift Valley and Tethysbaena from Oman was 20.13 MYA. The node of the most recent common ancestor of T. relicta, Tethysbaena from Oman, and the Tethysbaena from the Dominican Republic had a mean estimate of 35.84 MYA. The mean age for the node linking the Arabian-Caribbean clade (T. relicta + Tethysbaena sp. Oman + Tethysbaena sp. Dominican Republic) with the Mediterranean-Atlantic clade (T. scabra + T. leyoderi + T. ophelicola + T. argentarii + T. atlantomaroccana) was 40.42 MYA. The estimate for the root node linking Tethysbaena and Halosbaena was 79.96 MYA.

Table 3 Divergence times for Tethybaena species as estimated by the Bayesian evolutionary analysis method calculated using the COI gene molecular evolution based on Cánovas et al. (2016) and Bauzà-Ribot et al. (2012).

Node ages and highest posterior density ( ±95% HPD) ranges are given in million years round.

	Clade divergence (nodes)	Node age (MYA) (95% HPD range)	Geological period	
1	T. scabra—T. leyoderi	8.31 (10.15–3.97)	Miocene	
2	T. scabra + T. leyoderi—T. ophelicola	9.46 (14.20–5.71)	Miocene	
3	T. scabra + T. leyoderi + T. ophelicola—T. argentarii	10.71 (16.27–6.04)	Miocene	
4	T. scabra + T. leyoderi + T. ophelicola + T. argentarii— T. atlantomaroccana	32.41 (47.53–18.37)	Oligocene	
5	T. relicta—Tethysbaena sp. (Oman)	20.13 (41.69–13.25)	Miocene	
6	T. relicta + Tethysbaena sp. (Oman)—Tethysbaena sp. (Dominican Republic)	35.83 (51.41–22.16)	Eocene	
7	T. scabra + T. leyoderi + T. ophelicola + T. argentarii + T. atlantomaroccana—T. relicta + Tethysbaena sp. (Oman) + Tethysbaena sp. (Dominican Republic)	40.42 (56.09–25.72)	Eocene	
8	Tethysbaena—Halosbaena	79.96 (137.8–32.68)	Cretaceous	

Figure 5 Tethysbaena time tree using the COI gene.

A relaxed MCMC clock using the uncorrelated log-normal model and substitution rate based on Cánovas et al. (2016) were implemented in BEAST v2.4. Mean ages are presented on the nodes, and the 95% HPD (highest posterior density) are presented by the blue bars.

Discussion

In his monography on Thermosbaenacea, Wagner (1994) divided the Monodellidae family to two genera, the monotypic Monodella and the speciose Tethysbaena, which he named after the ancient Tethys Sea and the Greek word “βαινɛιν” (meaning “to walk”), referring to these animals as “walkers of the Tethys Sea”. He noted that although there is a great similarity among the different species, six species-groups can be identified based on morphological characters. With the later finding of T. exigua from Southern France, a seventh group was established (Wagner & Bou, 2021). Here, we analyzed the phylogenetic relatedness and divergence times of the two Levantine Tethysbaena species found in Israel: T. relicta from the Dead Sea-Jordan Rift Valley, and T. ophelicola, from the Ayyalon-Nesher-Ramla cave complex in central Israel.

According to Wagner (2012) and Wagner & Van Damme (2021), both Levantine species belong to “T. relicta-group” (together with four species from Oman, one species from Somalia and one species from Yemen), implying that these are sister taxa sharing a most recent common ancestor. Our phylogenetic results, based on the mitochondrial COI and the nuclear 28S rRNA genes, reject the morphology-based cladistics. The COI-based phylogeny supports the hypothesis suggesting that T. relicta shared an ancestor with Tethysbaena species from Oman and Dominican Republic, whereas the circum-Mediterranean species (including T. ophelicola) share another ancestor. Indeed, discrepancies between morphological cladistics and molecular phylogeny are common in cave fauna and were often attributed to their convergent troglomorphic traits (Bishop & Iliffe, 2012; Juan et al., 2010; Liu et al., 2017; Porter, 2007; Sendra et al., 2021).

Three paradigms determining the origin of the Thermosbaenacea and other phyla of subterranean crustaceans represented in the Dead Sea-Jordan Rift Valley and around the Mediterranean were defined. The earlier paradigm suggested that the Levantine Tethysbaena, among other subterranean salt-water fauna, have resulted from a late Pliocenic pre-glacial (Piacenzian, 3.600–2.588 MYA) marine transgression (Fryer, 1964; Hubault, 1937; Por, 1963). A narrow gulf penetrated into the coastal line near the present-day mount Carmel and then bent southwards along the Dead Sea-Jordan Rift Valley reaching a basin that extended south of the present Dead Sea (Picard, 1943). According to this paradigm, the Pliocenic Mediterranean was still inhabited by a very large number of Tethys remnants, including thermosbaenaceans, that were stranded in the Rift Valley and around the Mediterranean.

Por (1986) rejected the first paradigm, noting that the Pliocenic Mediterranean no longer contained the tropical fauna, including the Tethysbaena ancestor, and that the short-lived Pliocene transgression did not establish viable marine environments. Instead, he posited that these species represent marine fauna colonized during a Miocenic transgression (16–10 MYA) (Bar & Zilberman, 2016; Buchbinder & Zilberman, 1997), the last time that tropical sea penetrated inland in the Levant, and left stranded following a late Miocene regression (6–5.3 MYA) (Dimentman & Por, 1991; Por, 1987; Por, 1989). This second paradigm was supported by Guy-Haim et al. (2018) who applied a molecular clock approach to estimate the divergence time of the co-occurring caridean prawn Typhlocaris. Their assessment was based on a calibration node inferred from the end of the marine connection between the Mediterranean Sea and the Dead Sea-Jordan Rift Valley, marked by the top of Bira formation, dated to 7 MYA (Rozenbaum et al., 2016). They inferred a divergence time of Typholocaris from Ayyalon cave and Italy of 5.7 (4.4–6.9) MYA, at the time of the Messinian Salinity Crisis. During this event, the African plate moved towards the Euro-Asian plate, closing the Straits of Gibraltar and temporarily isolating the Mediterranean Sea from the Atlantic Ocean (Garcia-Castellanos & Villaseñor, 2011; Krijgsman et al., 1999). As a result, the Mediterranean Sea partly desiccated and transformed into small hypersaline basins, losing almost all its Miocenic tropical fauna, including those able to colonize subterranean waters (Carnevale et al., 2019; Delić et al., 2020; Por, 1987; Por, 1989).

With the discovery of the Ayyalon cave system and its endemic stygofauna in 2006, a third paradigm known as “the Ophel Paradigm” was developed by Por (2007). He identified the “Ophel” as a continental subterranean biome, subsisting on chemoautotrophic bacterial food, independently of the exclusive allochthonous epigean food of photoautotrophic origin. Within this biome, Tethysbaena are primary consumers, presenting a typical feeding behavior of upside-down swimming-gathering of sulfur bacteria or bacterial mats (Por, 2011; Wagner, 2012). Following the development of the new chemosynthetic-based biome paradigm, Por presented an alternative to the Tethys stranding paradigm, stating that the “Ophel paradigm falsified first of all my own, previously held views” on the diversification of the subterranean fauna in the Levant (Por, 2011). He noted that the pre-Messinian fauna of the fossiliferous taxa of the foraminiferans, the mollusks and the teleost fishes was similar to the recent Red Sea fauna or different only at the species level, and there is no indication for extinction of crustaceans during the Tertiary, thus inferring that the origin of the subterranean Levantine fauna is of earlier origin (Por, 2010). Por suggested that the Ophelic biome is possibly at least as old as the Cambrian, which had a diverse aquatic crustacean and arthropodan palaeofauna, including Thermosbaenacea (Por, 2011).

Cánovas et al. (2016) assessed the divergence time of the Western Mediterranean Tethysbaena, T. scabra from the Balearic Islands and T. argentarii from Italy using the COI gene. They based the nucleotide substitution rates on the mean rate estimated for a co-occurring anchialine stygobiont amphipod Metacrangonyx longipes, 1.32% per lineage and million years (95% CI [0.89–1.95]) (Bauzà-Ribot et al., 2012) and estimated the divergence time of T. scabra and T. argentarii to the early Tortonian, 10.7 MYA. Following Cánovas et al. (2016), we have used the COI gene to assess the divergence times of the Levantine Tethysbaena, T. relicta and T. opehlicola, and additional Tethysbaena species from around the Mediterranean, Arabian, and Caribbean Sea, using the Australian Halosbaena as an outgroup. Although the evolutionary rate used for calculating the Tethysbaena divergence closely matches the typical arthropod protein-coding mitochondrial substitution rates (1.15%/MY, Brower, 1994), it should be noted that the results based on this rate estimation should be treated with caution, as Metacrangonyx and Tethysbaena are only distantly related, and rates of molecular evolution tend to vary across genes, lineages and timescales (Hipsley & Müller, 2014).

Our analysis shows that the divergence times of Tethysbaena species are earlier than those of Typhlocaris species, pre-dating the upper-Miocene Messinian Salinity Crisis. Most divergence events occurred in the Miocene and Oligocene. The Dead Sea-Jordan Rift Valley T. relicta shares a most recent common ancestor with Tethysbaena from the Arabian Sea (Oman), dated to the early Miocene, 20.13 MYA (with 95% HPD of 41.69–13.25), corresponding with the Oligo-Miocene rift-flank uplift of the Arabian plate during the formation of the Red Sea and Gulf of Aden (34-20 MYA) (Omar & Steckler, 1995; Stern & Johnson, 2010). Both T. relicta and the Tethysbaena from Oman separated from the Caribbean Tethysbaena during the Eocene-Oligocene transition (38-30 MYA), when global cooling and tectonic uplift caused sea level decline and led to the establishment of the modern Caribbean Seaway (Iturralde-Vinent & MacPhee, 1999; Iturralde-Vinent, 2006; Weaver et al., 2016).

The most recent common ancestor of the Mediterranean Tethysbaena species—T. ophelicola from the coastal plain of Israel, T. scabra from the Balearic Islands, T. ledoyeri from Southern France, and T. argentarii from Italy—dated to the Tortonian in the Mid Miocene, 10.71 MYA (with 95% HPD of 6.27–6.04) as was previously found by Cánovas et al. (2016). The Ayyalon cave Tethysbaena, T. ophelicola, separated from other Mesogean (emerging Mediterranean) species around that time, 9.46 MYA (with 95% HPD of 14.20–5.71). The thermal water of the Ayyalon cave complex is part of the Yarkon-Taninim aquifer (Weinberger et al., 1994). During Oligocene-Miocene regressions (28-6 MYA), canyons were entrenched along the Mediterranean Sea shoreline, serving as major outlets of the Yarkon-Taninim aquifer, potentially forming anchialine karst caves (Frumkin, Dimentman & Naaman, 2022; Laskow et al., 2011). The ancestral habitat of Thermosbaenacea was first suggested to be of shallow marine origin (Fryer, 1964), stranded in the interstitial or inland groundwaters during periods of marine regression (Por, 1963; Stock, 1976; Tsurnamal & Por, 1971). An alternative hypothesis was proposed by Iliffe et al. (1984), who suggested that these stygobionts originate in the deep sea. This was later criticized by Stock (1986), among other reasons, because Thermosbaenacea colonizing marine anchialine caves have no relatives in the deep sea, but exhibit closer affinities to littoral interstitial species. Page et al. (2016) hypothesized that the ancestral habitats of Thermosbaenacea are Tethyan anchialine caves. Following our findings, we assume that the ancestor of T. ophelicola inhabited coastal anchialine caves in the Miocenic Tethys.

The most recent common ancestor of the Mediterranean and the Arabian-Caribbean clades of Tethysbaena is dated to the upper Eocene (40.42 MYA). During that period, the collision between the Arabian Plate and the Eurasian Plate resulted in the uplift of the Zagros Mountains in Iran (Mouthereau, Lacombe & Vergés, 2012). These mountain ranges acted as barriers, further isolating the Arabian Sea from the Mediterranean region (Sanmartín, 2003). The oldest, root node (Tethysbaena-Halosbaena) dated to 79.96 MYA (with the caveat of a low posterior probability and a large 95% HPD interval, 137.8–32.68 MYA). Page et al. (2016) established the phylogeny and divergence dates of the thermosbaeancean Halosbaena. They used the Tethysbaena-Halosbaena divergence as a calibration node, based on the presence of a continuous band of ocean crust through the length of the North Atlantic, indicating the maximum extent of the Tethys and the final opening of the Atlantic, dated to 107.5 MYA (with 95% HPD of 125–90). Thus, Tethysbaena ancestor in both our analysis and in Page et al. (2016) dates to the Cretaceous. The validity of the Paleozoic Ophel-driven hypothesis is also undermined by the deep phylogeny of peracaridean orders based on the small-subunit (SSU) rRNA gene, which showed that the thermosbaenacean lineage does not occupy a basal position relative to other peracarids (Spears et al., 2005). Numerous terrestrial sites having rich sulfidic waters have been discovered in the last century, and also recently (Nicolosi et al., 2022; Popa et al., 2019), after Por (2007); Por (2011) presented his “Ophel paradigm”. In general, we don’t see much evidence for a terrestrial Ophelic biome as old as the Cambrian. Rather, biogeographic, molecular, and hydrogeologic studies indicate that Cenozoic stranding events seem to be responsible for much of the observed subterranean fauna in these habitats (e.g., Delić et al., 2020; Esmaeili-Rineh et al., 2015; Frumkin, Dimentman & Naaman, 2022; Guy-Haim et al., 2018; Page et al., 2016).

Overall, the molecular clock-based divergence patterns presented here do not support the previously proposed hypotheses regarding the origins of the Levantine Tethysbaena. Instead, we infer a complex, two-stage colonization pattern of the Tethysbaena species in the Levant: (1) a late Oligocene transgression event, through a marine gulf extending from the Arabian Sea in the East to the Sea of Galilea in the west, leading to the colonization of T. relicta in the Dead Sea-Jordan Rift Valley, and (2) T. ophelicola, originating from the Mesogean Sea ancestor, inhabited anchialine caves in the coastal plain of Israel during the Mid-Miocene. Our results also show that the Cretaceous Tethysbaena ancestor first established in present-day Morocco, and then diverged into two groups. The first is a Tethyan group including Oman, the Dead Sea-Jordan Rift Valley and the Caribbean Sea. The second group formed around the emerging Mediterranean Sea, in its marginal aquifers, including Ayyalon, Southern France, Italy and the Balearic Islands.

Conclusions

Our multi-marker phylogenetic results reject the morphology-based cladistics and suggest that T. relicta shared a most recent common ancestor with Tethysbaena species from Oman and Dominican Republic, whereas the circum-Mediterranean species, including T. ophelicola, shared another ancestor. The molecular dating analysis suggests a two-stage colonization of the Tethysbaena species in the Levant, explaining their distant origins: a late Oligocene transgression leading to the colonization of T. relicta in the Dead Sea-Jordan Rift Valley, and a Miocene transgression in the Mediterranean region followed by a marine regression, stranding T. ophelicola in the coastal plain of Israel. The speciose Tethysbaena provides an exquisite opportunity for testing paleogeographic paradigms. Here we analyzed the phylogenetic relationships and divergence of nine out of twenty-seven known Tethysbaena species using the mitochondrial COI gene and the nuclear 28S rRNA gene. Future studies should examine additional species utilizing more genes or complete genomes to further unveil the phylogeny and biogeography of this unique and understudied group of ancient subterranean crustaceans.

The study of these subterranean species is not only an opportunity to broaden our understanding of paleogeography. It is also paramount for the protection of the hidden biodiversity found in these largely inaccessible habitats, which is nonetheless increasingly affected by human activity. Extraction of groundwater for irrigation and other uses, pollution, as well as quarrying, mining, and above-ground development may put these underground ecosystems at severe risk. The unique and often endemic nature of stygobiont species makes them even more prone to extinction, and extensive exploration of this under-explored biome, worldwide, is necessary in order to gain understanding and appreciation of the hidden biodiversity underground –an understanding that may pave the way for conservation of these species and their ecosystems.

Supplemental Information

Supplemental Information 1 28S rRNA sequences of Tethysbaena relicta and T. ophelicola

Click here for additional data file.

Supplemental Information 2 COI sequences with corresponding GenBank accession numbers of Tethysbaena ophelicola (Ayyalon) and Tethysbaena relicta (Dead Sea Rift Valley) used in the phylogenetic analysis

Click here for additional data file.

We are grateful to Boaz Langford, Israel Naaman, Yoav Negev, Lior Enmar, Ilia Kutuzov, Shlomit Cooper-Frumkin, Amitai Cooper, and the Israel Cave Research Center team for their support in field sampling. We thank Chanan Dimentman for kindly sharing his invaluable knowledge on subterranean Levantine fauna, and Stas Malavin for providing helpful comments on the draft. We are also thankful to Teo Delić for his constructive comments on the manuscript.

Additional Information and Declarations

Competing Interests

Author Contributions

Field Study Permissions

Data Availability

Tamar Guy-Haim is an Academic Editor for PeerJ.

Tamar Guy-Haim conceived and designed the experiments, performed the experiments, analyzed the data, prepared figures and/or tables, authored or reviewed drafts of the article, and approved the final draft.

Oren Kolodny conceived and designed the experiments, authored or reviewed drafts of the article, and approved the final draft.

Amos Frumkin conceived and designed the experiments, authored or reviewed drafts of the article, and approved the final draft.

Yair Achituv conceived and designed the experiments, authored or reviewed drafts of the article, and approved the final draft.

Ximena Velasquez conceived and designed the experiments, performed the experiments, authored or reviewed drafts of the article, and approved the final draft.

Arseniy R. Morov conceived and designed the experiments, performed the experiments, analyzed the data, prepared figures and/or tables, authored or reviewed drafts of the article, and approved the final draft.

The following information was supplied relating to field study approvals (i.e., approving body and any reference numbers):

No permit is required for the collection of Tethysbaena in the sampling sites.

The following information was supplied regarding data availability:

The data are available at NCBI GenBank: OR189199 –OR189204.

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
