# Peer review of "Shedding light on the Ophel biome: the trans-Tethyan phylogeography of the sulfide shrimp Tethysbaena (Peracarida: Thermosbaenacea) in the Levant"

_PeerJ, doi:10.7717/peerj.16690_

## Round 0.1 · original submission · Major Revisions

It can be extremely difficult to study small endemic subterranean species and we tend to know very little about these groups. I found the manuscript to be overall well written with a clear message. However, I did notice a few methodological problems that need to be addressed. While there is nothing wrong with using a singular gene (having studied subterranean beetles, I know it can be hard to extract enough DNA to do too many things with), it is important to comment on the limitations - especially given that it is CO1 (mitochondrial, non-recombining etc.) that you are using. In the age of SNP chips and whole genome sequencing, we know a lot more about what CO1 can provide and where it fails. Please talk about these failings in the context of your results and interpretations. If at all possible, data mining might be able to produce other genes or gene fragments that may be able to boost results (one of the reviewers took a look and this is possible to do). As a bit of a red flag, I also noticed that you used G-blocks and wonder why it was needed. The main advantage of using protein coding genes is that they align quite well. I wonder if this can be discussed further and/or possibly revisit gene alignments to verify accurate alignment. The usage of this program makes me slightly unsure of how you may have aligned your fragments together, and therefore, casts doubt on your results. Once these issues are addressed, this paper will fit nicely within the scope of PeerJ.

Reviewer 1 ·

Basic reporting

The study examines the current hypothesis of the origin of the Levantine Tethysbaena species using already published plus novel molecular data. Overall, the manuscript is well-written and easy to follow, including a good number of references that provide relevant information on the current state of the art. The manuscript follows the standard structure required by the journal and all figures and tables are clear and necessary to the manuscript. However, please clarify the inconsistency in Table 1 regarding accession numbers, as Cánovas et al (2016) provide accession numbers starting with “LN”, while Table 1 uses “CUS”.

Experimental design

The manuscript falls within the scope of the journal, presenting well-defined questions and a suite of methods that served their purpose. However, there are several issues that need to be addressed before acceptance.
The primary issue concerns the data itself. The study relies on a limited set of species for which only COI data is used (lines 128-133). The authors justify this decision by the low amount of variability detected by Cánovas et al (2016). However, since their focus differs from the one in the present manuscript (cryptic lineages vs phylogeographic of multiple species), it’s recommended to consider incorporating 28S data (or any other nuclear gene), which, despite low variability, could be useful for your manuscript’s purpose. I have retrieved all the sequences available for 28S and differences between species are around 4-5%.
Additionally, some other problems were detected:
- Lines 154-155: The use of GBlocks for COI sequences appears unnecessary, as COI sequences are straightforward and don’t contain ambiguous positions. I have retrieved all the COI data available for Tethysbaena and indeed, there is no reason to use the software. Can you explain why you used GBlocks for COI data?
- Results regarding Barcode: The authors stated that the DNA barcode consists of a fragment of 708 bp. However, the primers used amplify a region of 658 bp. I could check that some available sequences incorporate the primer region, that should be trimmed. Please, check the alignments and trim them if necessary.
- Bayesian analyses: The manuscript incorporates two Bayesian analyses, 1) MrBayes and 2) BEAST. These analyses may be complementary, but there is no reason to incorporate both. Moreover, there is no information on which posterior probabilities are used for Figure 3. I suggest removing MrBayes, as it is redundant.
- BEAST analysis: The authors use a 10% burn-in before combining the three independent analyses, but they should check for convergence and effective sampling-size to select the most appropriate threshold first. Also, the authors use a calibration considering rates for another amphipod species (lines 173-176) but in their work on Typhlocaris, the authors used a calibration based on a geological event. Could this be used for Tethysbaena?

Validity of the findings

I am afraid that relying solely on COI data for inferring a calibrated phylogeny may not be sufficient. As mentioned above, authors should explore the inclusion of 28S data or other nuclear markers, as mtDNA phylogenetics has known limitations. Multimarker approaches can be a great solution.
As a result, the unsupported nodes and wide HPD intervals in Figure 4 don’t allow to infer any clear pattern without any biases. I recommend the inclusion of at least one nuclear marker.

Additional comments

- Lines 87-95: Simplify the details of Typhlocaris to include only essential information (i.e., regions/countries).
- Lines 98-102: Provide a detailed explanation of the three hypotheses for clarity. They are explained in the discussion, but I think it will be necessary to include a better explanation of the hypothesis in the introduction.
- Lines 190-206: The morphological identification results are interesting but out of context in this manuscript. There is no need to provide such information.
- Lines 228-244: Avoid redundant information in the text (already presented in Table 2). There is no need to write a long list of nodes with their estimated age and HPD.
- Figure 4: Include only Periods and Epochs in the geological scale for improved readability

·

Basic reporting

Dear authors,
your paper presents a very interesting insight into biogegraphical development of the whole Circum-Mediterranean, or even wider, areas. As a speleobiologist I really liked its background and the premises from which you built your paper. However, I have to state that some parts of the manuscript are poorly resolved.
The major drawbacks are the discrepancies between the input data used and the results implying from them. There is not a single critique on behalf using only a single, mitochondrial gene in the analysis. In addition, the whole biome of Ophel did not receive any second thought about it existents, neither positive nor negative. It simply exists, although its existence is a bit ephemeral, and is very unclear (already in Por's original paper) how its elements are interconnected.
In addition to all mentioned, the reference parts are, in some parts, scarce. It seems, like the authors used only the first reference in 'google scholar' lists, without giving it a proper critical overview.
However, I think there is enough space to improve the paper by simplifying it and rephrasing/rewriting some parts of it.

Experimental design

While reading I bumped into serious problems' with the experimental desing. Herein I stressed out the major drawback, whereas most of them are included as "track changes" in the attached MS Wors document.
Despite dealing with the COI alignment, there are some major problems with the methods used to infer phylogenies. First, vast majority of sequences, listed in Supp. Material are referring to protein sequences and not gene sequences. However, this is easy-to-solve problem. The second part of my critique refers to usage of the alignment. I do not understand the background of usage "G-block" software. The software is usually used to cut badly aligned section. In COI, such sections should not exist. Simply because the gene is protein coding, therefore, no "holes" should be present. Finally, I conclude that either you did something wrong or the data is inappropriate.

Validity of the findings

The amount of the findings is not high, observing from the standards of the nowadays science, but is highly important for the small, poorly known group of subterranean animals. Moreover, the distributional range covers half of the globe, therefore I strongly encourage publishing the presented work. However, and first of all, large interventions in the extant version of the manuscript are needed before doing this.

Additional comments

Finally, as a reviewer I made many comments. I hope they will not be understood as being negative, but and in the first line, as a critical overview with a wish to improve your manuscript.
Best wishes
Teo Delić

·

Basic reporting

A nicely put together paper with some interesting molecular results. I have suggested some corrections to the text and they are marked in the manuscript.
References seem sufficient. Context for the project was presented well. The authors presented their results without inflating the importance of the work or drawing too much from the results.
A nice CO1 snapshot into relationships between the thermosbaenaceans in the region.

Experimental design

They present a small CO1 phylogeny with some dating to look at several existing origin hypotheses. The work illustrates nicely how cryptic morphology can lead to ambiguous or cotentious evolutionary hypotheses.
I did worry that the type locality is now lost for T. relicta and that the authors have not discussed how they might have been able to properly identify the species (DNA from type specimens?). Perhaps sampling at a point that is 6km from the type loaclity is sufficient but it worries me that this wasn't more fully detailed.

Validity of the findings

conclusions are well stated. I am happy for this paper to be published with only minor revision.

---

## Round 0.2 · accepted · Accept

The authors have done a great job at addressing the concerns and comments of reviewers. The addition of a second marker strengthens the results immensely and also removes all the concerns that arise with using only the CO1 marker. Authors have also realigned data without GBlocks software, which had originally been a red flag for most reviewers and myself, as well as re-wrote sections containing confusing/unsatisfactory text. I have assessed this manuscript and am happy with the current version. This manuscript is ready for publication.